# Immunomorphogenesis in Degenerative Disc Disease: The Role of Proinflammatory Cytokines and Angiogenesis Factors

**DOI:** 10.3390/biomedicines11082184

**Published:** 2023-08-03

**Authors:** Natalya G. Pravdyuk, Anna V. Novikova, Nadezhda A. Shostak, Anastasiia A. Buianova, Raisa T. Tairova, Olga I. Patsap, Aleksandr P. Raksha, Vitaliy T. Timofeyev, Victor M. Feniksov, Dmitriy A. Nikolayev, Ilya V. Senko

**Affiliations:** 1Acad. A. I. Nesterov Department of Faculty Therapy, Pirogov Russian National Research Medical University, Ostrovityanova Str., 1, 117997 Moscow, Russia; annove2008@mail.ru (A.V.N.);; 2Center for Precision Genome Editing and Genetic Technologies for Biomedicine, Pirogov Russian National Research Medical University, Ostrovityanova Str., 1, p. 1, 117513 Moscow, Russia; anastasiiabuianova97@gmail.com; 3Federal Center of Brain Research and Neurotechnologies FMBA, Ostrovityanova Str., 1, p. 10, 117513 Moscow, Russia; cleosnake@yandex.ru (O.I.P.);; 4Pirogov City Clinical Hospital No. 1, Moscow Healthcare Department, Leninskiy Prospekt, 8, 117049 Moscow, Russia

**Keywords:** back pain, young age, degenerative disc disease, proinflammatory cytokines, Angiogenesis markers, CD31, chondrocyte clusters, immunohistochemistry, Modic changes

## Abstract

Back pain (BP) due to degenerative disc disease (DDD) is a severe, often disabling condition. The aim of this study was to determine the association between the expression level of proinflammatory cytokines (IL-1β, IL-6, and IL-17), angiogenesis markers (VEGF-A and CD31) in intervertebral disc (IVD) tissue and IVD degeneration in young people with discogenic BP. In patients who underwent discectomy for a disc herniation, a clinical examination, magnetic resonance imaging of the lumbar spine, histological and immunohistochemical analyses of these factors in IVD were performed in comparison with the parameters of healthy group samples (controls). Histology image analysis of IVD fragments of the DDD group detected zones of inflammatory infiltration, combined with vascularization, the presence of granulation tissue and clusters of chondrocytes in the tissue of nucleus pulposus (NP). Statistically significant increased expression of IL-1β, IL-6, IL-17, VEGF-A and CD31 was evident in the samples of the DDD group compared with the controls, that showed a strong correlation with the histological disc degeneration stage. Our results denote an immunoinflammatory potential of chondrocytes and demonstrates their altered morphogenetic properties, also NP cells may trigger the angiogenesis.

## 1. Introduction

The overwhelming number of back pain (BP) cases is due to a degenerative lesion of the spine, which is accompanied by a complex of changes in the vertebral–motor segment. Modern research in the pathogenesis of spinal degeneration covers all the proposed mechanisms: from molecular/cellular and genetic to psychosocial, which leads to the formation of a multimodal treatment approach for BP, especially during the BP chronization.

Comparative studies of the prevalence of chronic BP among the young contingent are few. According to European population studies using questionnaires, the one-year prevalence of adolescent BP reaches 83%. The official registration and report in the health care system on the presence of BP is carried out only in 0.01–12.5% of cases, of which the vast majority of people aged 15 to 24 years have functional disorders. [1]. BP associated with intervertebral disc (IVD) lesion in age group 15–24 years is much less common than in the older one and ranges from 0.5% to 6.8% of all treated hernias in patients of different ages [2]. The majority of operated patients (60%) had protrusions, 33%—subligamentous disc herniation, 9%—hernias with rupture of the annulus fibrosus (AF); the most frequent localizations were levels L4–L5 (54%) and L5–S1 (34%). With the initial existence of BP diagnosed at an earlier age period, further progression of pain is possible [3]. Previous studies have identified 2-fold increase in patient numbers with chronic back pain in 8 years (from 4.2% to 9.6%) [4]. It has been demonstrated that the longer the period of disability due to acute BP, the less likely it is to return to work. The chances of a patient suffering from BP with temporary disability for 6 months to return to work are approaching zero 2 years after the onset of the disease [5].

Over the past two decades, the study of degeneration of the IVD and adjacent vertebral bodies at the microlevel has focused the attention of researchers on the immunological mechanisms of cellular matrix damage. Thus, it has been shown that in the early stages of disc degeneration, there is a shift in the balance of anabolic and catabolic activity of extracellular matrix molecules and IVD chondrocytes, which occurs with the direct participation of proinflammatory cytokines. The early stage of disc degeneration—preclinical—consists in the disintegration of the intercellular substance, hypoxemia, acidification of the biochemical medium and accumulation of lactate [6], loss of hygroscopicity of glycosaminoglycans and chondroitin sulfates of the nucleus pulposus (NP) [7]. Fragmentation products include hyaluronic acid, biglican, versican, decorin, elastin and laminin, which aggravate the inflammatory cascade in the disc [8]. Moreover, inflammatory cytokines induce the expression of matrix-destroying proteins [9]. This leads to a decrease in the elasticity and surface tension of the disc NP with an increase in the recovery time of its height under axial loads, dehydration, and violation of the cellularity of already extremely small chondrocytes [10]. Induction of inflammatory activity in the disc and in adjacent structures leads to advanced stages of degeneration, protrusions, hernias and stenosis of the spinal canal [11].

Traditional inflammation, as is known, involves the formation of granulation tissue consisting of macrophages and T-lymphocytes. Recent findings have revealed that pro-inflammatory molecules involved in the development of spinal degeneration include interleukin (IL)-1, IL-2, IL-4, IL-10, IL-12, IL-15, IL-17, prostaglandin-E 2 (PGE2), interferon (IFN)-γ, whose expression increases locally, as well as IL-6, IL-8, tumor necrosis factor (TNF)-α, the expression of which increases in the blood serum [8]. It is known that macrophages are the most numerous of all inflammatory cells. After activation, they secrete cytokines and compounds that cause disc degeneration, and can even spread to nearby intact disc tissues in advanced stages of degenerative disc disease (DDD) [12]. This implies an active migration of immunocytes to the lesion/inflammation site along the vascular bed. However, the initial signs of degeneration are detected in discs that have retained their anatomical integrity, i.e., in the absence of obvious mechanical micro- and macro-injuries that could act as a “window of migration” of immunocompetent cells with the launch of inflammation. In view of this, it remains unclear whether macrophages play a dominant role in disc degeneration by analogy with other tissues, or the initiation of inflammation begins in their absence due to the transformation of the cells of the NP and the AF themselves, and what is the role of disc vascularization with loss of its immunological tolerance and repair. It is likely that inflammatory cytokines control the degenerative process in the disc, participate in the mechanisms of pain and are the most relevant target for pathogenetic therapy [13].

The aim of our study was to identify the expression level of proinflammatory cytokines (IL-1β, IL-6, and IL-17) and markers of angiogenesis (vascular endothelial growth factor (VEGF)-A and CD31) in IVD tissue and their correlation between tissue degeneration and indicators of discogenic BP in young people.

## 2. Materials and Methods

### 2.1. Patients

This study was approved by the Ethics Committee of Pirogov Russian National Research Medical University (protocol No. 208 dated 17 May 2021). A total of 34 patients (17 men and 17 women) were sequentially selected in the neurosurgical department of the N.I. State Clinical Hospital No. 1. Pirogov. They were scheduled to have a microdiscectomy due to progressive BP and/or neurological symptoms associated with a hernia. The inclusion criteria in the DDD group were: young age (median age—32.00 years [29.00–39.50]) and BP associated with IVD hernia confirmed by instrumental data (magnetic resonance imaging (MRI)). Exclusion criteria were: spinal injury at the time of this study and in the anamnesis, tumor, infections of the spine and other organs, inflammatory spondyloarthritis, surgical interventions in the previous 30 days. The intensity of BP was clinically assessed according to VAS: the types of pain were acute or chronic (according to the classification of The International Association for the Study of Pain (IAPS) 2021), the pain lasts years (both persistent and recurrent) and the functional limitation of spinal mobility was scored by the Backache Index. All patients underwent MRI without contrast agent and anesthesia on a Toshiba Excelart Vantage Atlas-X 1.5 tesla device in three sections (sagittal T2-weighted, coronary T2-weighted short-tau inversion-recovery and axial T2-weighted; slice thickness = 1.5 mm), and scan time was 15 min. The specific absorption rate level was below 2 W/kg for the body, magnetic field strengths reached 1.5 T and gradient strengths up to 30 mT/m with a peak slew rate of 130 T/m/s.

### 2.2. Tissue Samples

There were obtained 34 samples of disc tissue from patients of the DDD group during the discectomy and 7 samples of control disc tissue. The biopsy specimen consisted of multiple fragments of whitish, dense elastic tissue and cartilaginous density tissue with a total size of 6 × 5 × 1 cm. Control samples were obtained during spinal stabilization and removal of the dropped segments of intervertebral cartilage L4/L5, L5/S1 in 7 healthy individuals due to acute auto injury within 1 h after the incident. The age of the control group varied from 27 to 44 years. The patients had no history of diseases of the spine and musculoskeletal system, infectious or oncological diseases, and did not abuse narcotic drugs. Patients of both groups signed an informed consent to participate in this study.

### 2.3. Histological Examination

IVD tissue samples were fixed immediately after discectomy in 4% paraformaldehyde/phosphate-buffered saline (pH = 7.4) for 72 h in room temperature. In the presence of calcification or residual bone material, decalcification in HCl was performed, then samples were processed into paraffin wax. Four-micron sections from the tissue blocks were cut and stained with hematoxylin and eosin according to standard protocol.

### 2.4. Immunohistochemistry

All sections were examined by immunohistochemistry on the automated VENTANA BenchMark ULTRA platform. For investigating angiogenesis, a primary mouse anti-VEGF-A (1:500 dilution; Cloud-Clone Corp., Katy, TX, USA) and primary rabbit anti-CD31 (1:150 dilution; ab28364, Abcam, Cambridge, UK) were used. To identify processes of inflammation, we also applied primary mouse anti-IL-1β, anti-IL-6 and anti-IL-17 (1:150 dilution; Cloud-Clone Corp., USA).

### 2.5. Morphometric Analysis

All slides were visualized using a Axio Imager.Z2 microscope (Carl Zeiss, Jena, Germany). Cell counts were performed on all independent fields of photomicrographs captured with EC Plan-Neofluar 40× objective. The region imaged was NP. A positive signal was observed as dark brown/yellow brown coloring. Image analysis was carried out in the Fiji program. The stage of degeneration was established in accordance with the criteria of Sive [14], where 0–3 points were considered as the norm, 4–9—degeneration, and 10–12—severe degeneration. The markers expression was assessed by percentage ratio of immunopositive cells to the total number of cells.

### 2.6. MRI Data Analysis of the Lumbar Spine

According to the MRI data, the IVD height was assessed at the hernia level, the Pfirrmann DDD stage at the operated level (disappearance of the difference between NP and AF and a decrease in the intensity of the signal from the NP), the existence of reactive changes in the adjacent vertebral bodies according to the Modic type (1, 2, 3 types)—changes in the MR signal intensity from the bone marrow in the T2-weighted image and with fat suppression, the severity of the lesion Modic changes in area in the sagittal and/or coronary section and the having an erosive type of lesion of the vertebral body endplates (EP).

### 2.7. Statistical Analysis

Statistical data were analyzed using Graph Prism 8.0.1 (GraphPad, La Jolla, CA, USA). The distributions were determined to be parametric by Shapiro–Wilk testing. Data with a normal distribution were presented as the values of the mean ± SD, non-normal variables were presented as median [Q1, Q3]. To verify the differences between the expression of immunohistochemical markers and the severity of histological degeneration, the Mann–Whitney U test was used. Spearman’s criterion was used for correlation analysis. A *p*-value <  0.05 was considered statistically significant.

## 3. Results

### 3.1. Clinical Characteristics of Patients and MRI Data

BP characteristics in the DDD group are presented in Table 1. Pain intensity median corresponded to 62 [46.75–87.00] mm. Most of the patients (30 or 88.24%) suffered from chronic BP. At the same time, the recurrent type of pain was found in 73.53%, and persistent (permanent without a bright gap)—in 26.47%. The duration of the pain syndrome ranged from 0.1 to 25 years. A total of 28 patients (82.35%) had moderate and severe degrees of functional disorders according to the BAI index.

### 3.2. Results of MRI of the Lumbar Spine

Localization of IVD hernia in 23.53% (*n* = 8) of individuals was at the level of L4/5, in 76.47% (*n* = 26)—at L5/S1. The average value of the IVD height demonstrated a significant narrowing of the intervertebral space at the level of the operated disc (L4/5, L5/S1) compared to the L3/4 level—the average values of the IVD height were 0.86 ± 0.16 and 2.61 ± 0.15 mm, respectively (*p* < 0.01). The stage of IVD degeneration at the discectomy level according to Pfirrmann corresponded to the 4th and 5th stages in 55.88% (*n* = 19) and 38.24% (*n* = 13), respectively. In half of the patients (*n* = 19; 55.88%), reactive changes in the adjacent vertebrae bodies were detected: the bone marrow edema (Modic-1) in 26.47% (9 patients) and fatty degeneration of the bone marrow (Modic-2) in 29.41% (10 patients) (see Figure 1). Almost half of the patients in the DDD group (*n* = 16; 47.06%) had a combination of IVD hernia plus Modic changes with EP erosive lesion of the adjacent vertebral bodies.

### 3.3. Histological and Morphometric Data

In the IVD samples of the DDD group, inflammatory infiltration zones were identified, combined with vascularization (see Figure 2A) and the presence of granulation tissue. There were also the regions of hyaline cartilage with inflammatory cells infiltration.

The morphological degeneration stage in the samples of the control group was Me = 1.000 [0.000–3.000] points, while of the DDD group was Me = 7.000 [6.000–10.000] (*p* < 0.0001). In the samples of DDD patients, clusters of NP cells (clusters of chondrocytes) were detected (see Figure 2B), especially in damaged areas that were absent in the control samples, which can be regarded as another morphological sign of disc degeneration. The average area of chondrocyte cluster was 3645.393 ± 551.701 μm^2^ and the average number of cells in them was 9.411 ± 3.382.

### 3.4. Immunohistochemical Analysis

Immunohistochemical staining revealed a statistically significant difference in the expression levels of all the proinflammatory cytokines and vascular endothelial growth factors determined in all patients with BP and DDD compared with the control group (see Figure 3 and Figure 4, Table 2).

A high expression of IL-1β was revealed in the NP chondrocytes (DDD: 56.15 ± 24.40%, control: 16.43 ± 10.10%; *p* < 0.001) and around the vessels; medium expression in the disc matrix (*p* < 0.05) (see Figure 5). Expression of IL-1β was positively correlated with histological degeneration stage (r = 0.606, *p* < 0.0001). IL-1β was expressed in two samples of the control group.

The IL-6 expression was medium in the chondrocytes of the NP (DDD: 49.65 ± 21.41%, control: 29.86 ± 8.45%; *p* < 0.05) (see Figure 5). The correlation coefficient of 0.597 (*p* < 0.0001) indicated a moderate positive correlation between IL-6 expression and histological degeneration stage.

High IL-17 expression in NP cells (DDD: 63.71 ± 21.12%, control: 8.71 ± 5.77%; *p* < 0.0001) and around vessels (see Figure 5) states the appearance of this cytokine due to activation of NP chondrocytes and probably due to the migration of immune cells through newly formed disc vessels. In the DDD group, a coincidence of the spatial expression of IL-1β and IL-17 was found (in 21.62 ± 9.90% of immunopositive chondrocyte)–in the endothelium and in the perivascular zone, as well as in the vascular lumen (*p* < 0.01). It confirms the hypothesis that they act synergistically during inflammation processes. The positive correlation between IL-17 expression and histological degeneration stage was strong (r = 0.616, *p* < 0.0001). This marker was expressed around the vessels in two control samples, along the disc edges.

A high expression of VEGF-A was detected in clusters of cells (it occupies more than ¾ of the cluster) along with weak expression in the matrix and around single cells of NP (DDD: 74.68 ± 19.17%, control: 17.43 ± 7.74%; *p* < 0.0001) (see Figure 6), indicating that NP cells (as single, and in clusters) trigger the angiogenesis process. The moderate correlation between VEGF-A expression and histological degeneration stage was positive (r = 0.519, *p* = 0.001).

CD31 was found in 22 patients. A high expression of CD31 was detected around NP single cells and in the disc matrix, where were no vessels (see Figure 7). This marker was also presented in the endothelium of the vessels in granulation tissue in IVD samples of four patients. Cracks in the disc were not stained on CD31, which excludes the primary role of vascular ingrowth into IVD in the formation of IVD defects. CD31 was not expressed in any control sample.

## 4. Discussion

All patients were young and, regardless of gender, had intense pain syndrome associated with severe IVD degeneration according to MRI (4th and 5th stages of DDD according to Pfirrmann), a long-term hernia at the level of L4/5 or L5/S1, in half of cases associated with reactive spondylitis of adjacent vertebral bodies. A significant relationship between Pfirrmann radiological classification of DDD and Thompson morphological assessment system according to the literature data allowed us to use both assessment systems of IVD lesions in patients in our work [15].

The inflammatory pattern starts and supports the breakdown of the cartilage matrix of the vertebral–motor segment [16]. A decrease in the content of aggrecan triggers the process of vascularization of the disc and with the appearance of annular tear, the IVD loses its immunological stability: the activation of chondrocytes of the NP occurs and the involvement of immunocompetent CD-68 lymphocytes with phagocytic activity, macrophages and mast cells from the growing vessels. The expression of TLRs receptors on chondrocytes and fibroblasts sensitizes them to matrix degradation and activates them as immunologically active cells, that is, changes the immune phenotype of the IVD cells. Fragments of cleaved hyaluronic acid boost the expression of interleukins and matrix metalloproteinases by cells of the NP [17]. The secretion of TNF-α, IL-1α/β, IL-6 and IL-17 is triggered, which, in turn, activate matrix metalloproteinases, thrombospondin and disintegrin, aggravating the degradation of the IVD matrix, this results in a vicious circle of “degeneration—inflammation” in IVD [16]. Cytokines promote the production of chemokines (CCL2, CCL3, CCL5, CCL7, and CCL8), PGE2, inducible isoform of nitric oxide synthases and cyclooxygenase-2, that contributes to the further attraction of T and B cells, macrophages, neutrophils, mast cells, exacerbating the cascade of inflammatory reactions [16,18]. IL-1ß is a key cytokine in the process of IVD degeneration and associated BP [19], it induces an increase in the content of VEGF in the IVD under hypoxia [20,21], as well as the production of IL-6 and IL-17 [22].

We analyzed the expression level of IL-1β, IL-6, IL-17, vascular endothelial growth factor A and CD31 of the removed patients IVD tissue, and compared it with tissue samples of the group without BP. It is important that the expression level of all studied cytokines was positively correlated with the histological degeneration level. This does not leave any doubt about the pathogenic linkage between immune inflammation and severe degeneration, and also shown the cytokines primacy in the progression of structural disc disorganization [23].

The high expression of IL-1β, IL-6 и IL-17 in the chondrocyte clusters indicates the transformation of the immunomorphogenetic properties of the structural cells of NP and AF, which at one point begin to acquire immunogenic properties. The spatial synergism of IL-1β and IL-17 sheds light on their combined effect in disc degeneration, and allows us to consider their participation in reactive spondylitis and erosion of the vertebral bodies as the main promoters of bone remodeling in DDD by analogy with autoimmune rheumatic diseases (rheumatoid arthritis, psoriatic arthritis, etc.) [24,25].

A high VEGF-A expression level in the NP, which is normally a non-vascular structure, proves the angiogenic abilities of chondrocytes were long before the appearance of a vascular microgrid there. CD31 expression not only in the vascular endothelium, but also in the disc matrix, where vessels were absent, may indicate an increase in the permeability of the disc, since the marker is expressed on cells of the immune system. These data suggest angiogenesis in the discs in not so much due to vessel ingrowth from outside the AF but rather due to endotheliocytes in situ, which is confirmed by the data of previous studies in which chondrocytes expressed VEGF-A, VEGF-B and VEGF-C [26]. Thus, local VEGF inhibition at the onset of DDD (2–3 stages by Pfirrmann) may be considered as a potential target for inhibiting disc degeneration [27]. In our samples, the cracks of AF and NP were not stained with CD31, which denotes their mechanical genesis, not angiogenic, despite the blood cells detected in the lumen. This confirms the assumption of a transformation of the morphogenetic properties of chondrocytes in IVD under the influence of altered disc biomechanics. It is most likely that the start of inflammation in the discs can be caused by hyperphysiological stresses. The relationship between mechanical stress and inflammation in the peak load regions, as previously shown, is realized at the molecular level through the channels of transient receptor potential that exist in degenerated IVD. Thus, a locally altered load distribution can potentially affect the morphogenetic properties of fibroblasts through mechanotransduction [8].

EP’s involvement in the process of inflammation according to both histological and immunohistochemical analyses demonstrates the loss of immune privilege by the disc, the spread of immune inflammation beyond its limits to the adjacent areas of the vertebral bodies and the formation of reactive spondylitis and erosive lesions of the bone EP part [28], recorded on MRI images in patients with DDD and BP. So, Ma Kh. L. and co-authors revealed the expression of the same proinflammatory cytokines in vertebral bone marrow with Modic-1 as in the degenerated disc: IL-1, -6 and -8, -10; TNF-α, PGE2, chemotactic protein of monocytes-1, which demonstrates the prevalence of immune inflammation not limited to the border of the disc [29]. A cascade of immune inflammation penetrates disc from the bone marrow of the vertebral bodies through damaged EP. The probability of an inflammatory reaction in the bone in situ is low due to the fact that Modic changes are detected in patients exclusively in the presence of hernia and/or erosion of the EP, which has been demonstrated in similar studies [30,31]. In turn, the feedback of the EP bone part with the state of the NP cells, their activity and influence on the hydrophilic properties of the extracellular matrix has been demonstrated in studies on the nutritional function of the EP [32].

According to the literature, IL-1β significantly increased the expression of IL-6 and IL-8, VEGF, chemotactic factor (MCP-1) and disc degradation factor (MMP-3) in IVD cells during hypoxia [21]. The level of IL-6 was elevated both in the patients’ serum with DDD and spinal injuries, but the immunohistochemical expression of IL-6 was significantly higher in the degenerative disc tissue [33].

In addition, the *IL-6* gene increased expression in patients’ blood serum with BP and DDD correlated with inflammatory edema of the bone marrow of adjacent vertebral bodies (Modic-1) compared with individuals who had fatty transformation of vertebral bodies (Modic-2) [34], which point at an inflammatory pattern for all the vertebral–motor segment, and not just for the IVD. A “degenerative profile” with increased expression of VEGF, MMP-2 and MMP-3 was observed in elderly people in all parts of the spine and exceeded that in patients younger than 35 years [35]. VEGF was higher expressed in the elderly in all departments, and the level of IL-1β was higher in the lumbar discs, which does not contradict the results of our study.

Finally, the data of our study allow us to speak about the active immunomorphogenesis of chondrocytes as one of the key factors in the initiation and/or maintenance of disc degeneration. It is known that NP cells may have the properties of stem cells or progenitor cells in the IVD tissue [36] and also the properties of the notochordal cell subpopulation of chondroblasts characteristics for embryonic period of IVD formation [37]. A recent study on the heterogeneity of the phenotypes and functions of NP cells (using single-cell RNA sequencing of five patients) demonstrated the presence of three types of NP cells—chondrocytes (the vast majority of cells with three populations—cartilage progenitor cells, fibrochondrocyte progenitor cells and homeostatic chondrocytes), endotheliocytes and macrophages [38]. Based on the marker gene expression specific to the cell line, seven subclusters were identified among chondrocytes, differing in heterogeneity and biological functions. Thus, the C-1 and -3 subclusters of chondrocytes had an inflammatory phenotype. It is these cell lines that can be considerably activated during disc degeneration, which explains the transformation phenomenon of the NP cells immunomorphogenesis at the level of the entire population of chondrocytes, i.e., their pathological differentiation in DDD. Moreover, inflammatory subclusters of chondrocytes and macrophages, which presents in much smaller numbers in the NP, another subcluster is able to synthesize the angiogenesis factor and enhance the adhesion of endotheliocytes. Thus, individual immune phenotypes of degenerated disc chondrocytes act as conductors of the inflammatory cascade and vascularization in IVD.

In this study, low levels of expression of cytokines IL-1β, -6, -17, and VEGF-A were observed in the samples of the control group, which allows for the probability of the onset of disc «predegeneration», which did not manifest histologically and at the macro level (no sign of degeneration was detected visually and microscopically). This finding may indicate an early preclinical and pre-morphological stage of IVD degeneration, which can manifest itself by a displaced cytokine profile and altered mechanoelastic disc functions with pronounced axial loads, which requires further study.

This study included only those patients who were shown surgery due to the ineffectiveness of conservative therapy, which causes a shift in clinical and morphological indicators towards more severe data. We could not assess the disc degeneration by MRI in a group of healthy donors, conclusions about its absence or minimal manifestations are based only on the data of the pathomorphological findings.

## 5. Limitation of Our Study

The main limitation of this study was the small sample size. The control group was not strictly match with the DDD group in terms of relevant factors. The possible confounding effects of age, gender, body mass index, smoking status, comorbidity, medication use, and other factors were not taken into account. We did not perform any functional or molecular assays to confirm the causal relationship between cytokine expression and IVD degeneration, or to elucidate the underlying mechanisms of inflammation and angiogenesis in the disc. Additionally, the effect of steroids on disc cytokine levels has not been studied.

## 6. Conclusions

Our data from clinical and instrumental studies of patients with BP and DDD in combination with histological and immunohistochemical analyses of degenerated IVD tissues provided a three-dimensional picture of the “inflammatory-degenerative pattern” in the lumbar spine degeneration and the formation of IVD hernia. Immune inflammation in discs with IL-1β, -6, -17 was present in all discs of the DDD group and had a high level of correlation with the histological stage of IVD degeneration. This proves that local immune inflammation is an obligatory component of the degenerative cascade in the IVD, initiates angiogenesis in the NP, AF of discs with the expression of angiogenesis factors VEGF-A on the surface of chondrocytes and CD31 in the extracellular matrix, and extends to the EP. It is a key mechanism for the development of reactive osteitis of adjacent vertebrae with formation of inflammatory erosive lesions of the vertebral–motor segment in BP. The detection of cytokine expression on the surface of chondrocytes, including clusters of chondrocytes, demonstrates altered morphogenetic properties of chondrocytes with transformation into an inflammatory cell phenotype and the secondary role of macrophages in IVD tissue. The results obtained will help to identify molecular and cellular targets (IL-1β, -6, -17, and VEGF) and develop basic strategies for biological, cellular and targeted therapy of BP in young people in the early stages of DDD. In particular, we may try to use biopolymers that prevent neovascularization as a matrix model for intradiscal injection and as a scaffold for stem cell therapy.

## Figures and Tables

**Figure 1 biomedicines-11-02184-f001:**
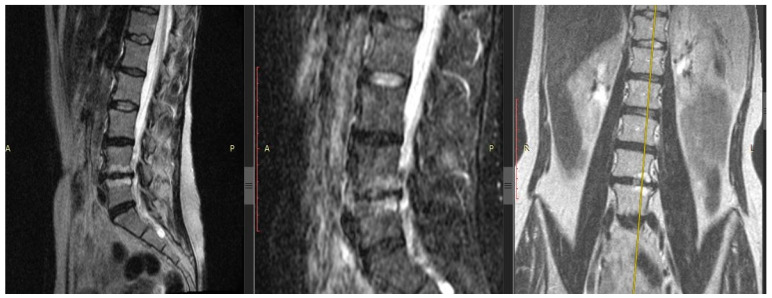
Patient with back pain. Magnetic resonance imaging of the lumbar spine, modes T2-weighted images with fat suppression in sagittal sections (left and center images, respectively), T2-weighted short-tau inversion-recovery image in the coronary section (right image). Right-sided lumbar scoliosis, the 5th stage of degenerative disc disease by Pfirrmann at the L4/5 and 4th stage—at the L5/S1 level, with IVD hernias L4/5 and L5/S1, erosion of the endplates and Modic changes type 1 in the vertebral bodies L4/5 (yellow axis carried out via Modic-1). Modic-1 = the bone marrow edema; Modic-2 = the bone marrow fatty degeneration.

**Figure 2 biomedicines-11-02184-f002:**
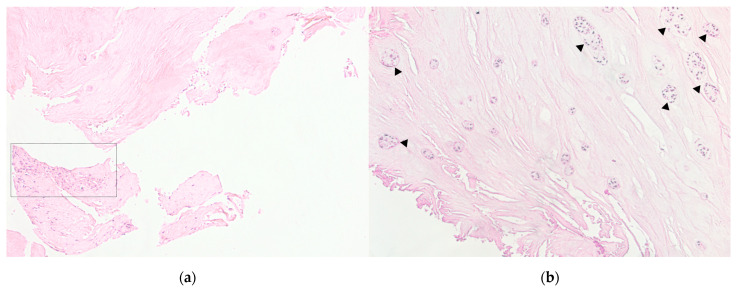
Patient intervertebral disc sample obtained as a result of microdiscectomy in the patient in her 40 s, with a hernia of the L5/S1 vertebrae. Own data, 2020. Hematoxylin–eosin staining. (**a**). Light microscopy, magnification 100×. Zones of inflammatory infiltration, combined with vascularization and granulation tissue. (**b**). The sample of the same patient, magnification 100×. Clusters of nucleus pulposus cells (triangles) are a characteristic sign of disc degeneration.

**Figure 3 biomedicines-11-02184-f003:**
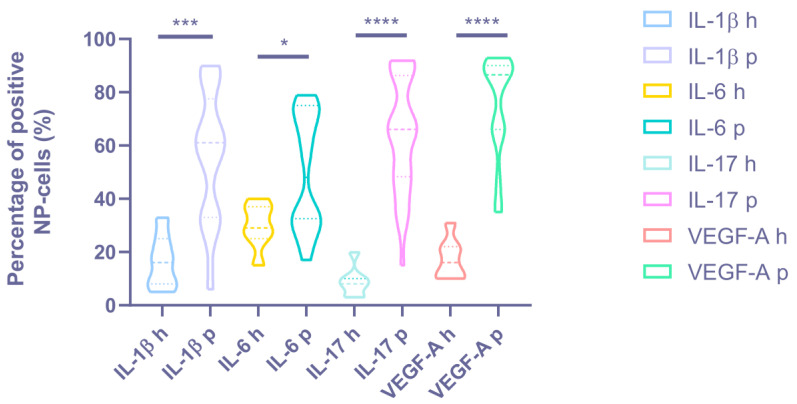
The percentage of positive NP cells in the intervertebral disc tissue of the degenerative disc disease (“p”—patients) and control group (“h”—healthy) discs. * *p* < 0.05. *** *p* < 0.001. **** *p* < 0.0001. NP = nucleus pulposus, IL = interleukin, and VEGF-A = vascular endothelial growth factor A.

**Figure 4 biomedicines-11-02184-f004:**
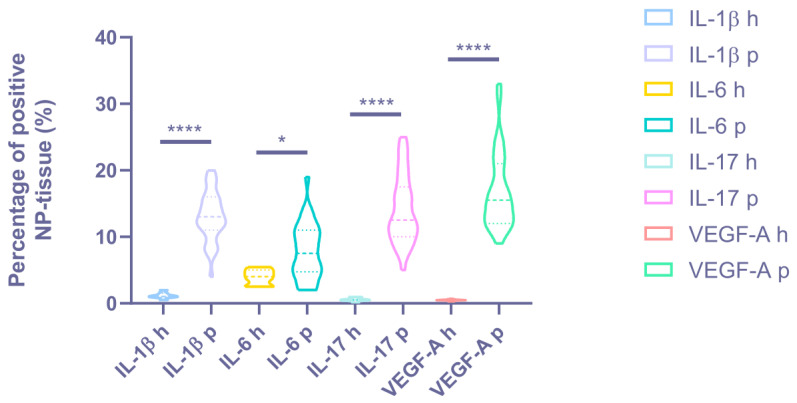
The cytokine expression degree in the intervertebral disc matrix of the degenerative disc disease (“p”—patients) and control groups (“h”—healthy). * *p* < 0.05. **** *p* < 0.0001. NP = nucleus pulposus, IL = interleukin, and VEGF-A = vascular endothelial growth factor A.

**Figure 5 biomedicines-11-02184-f005:**
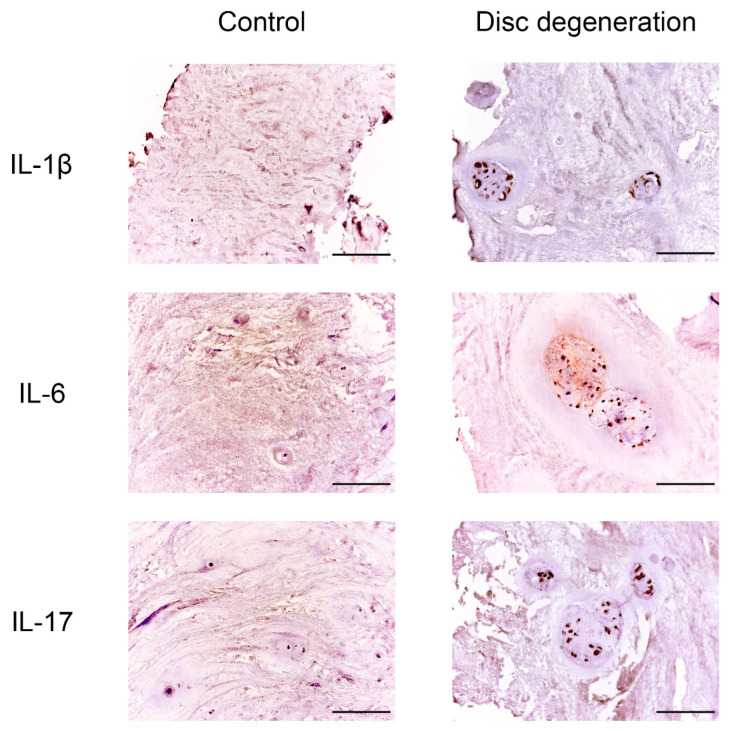
Expression of interleukin (IL)-1β, IL-6, and IL-17 in intervertebral disc tissue. Scale bars: 100 µm.

**Figure 6 biomedicines-11-02184-f006:**
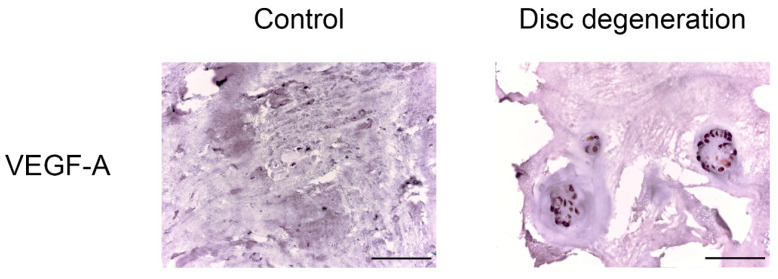
Vascular endothelial growth factor (VEGF)-A expression in intervertebral disc tissue. Scale bars: 100 µm.

**Figure 7 biomedicines-11-02184-f007:**
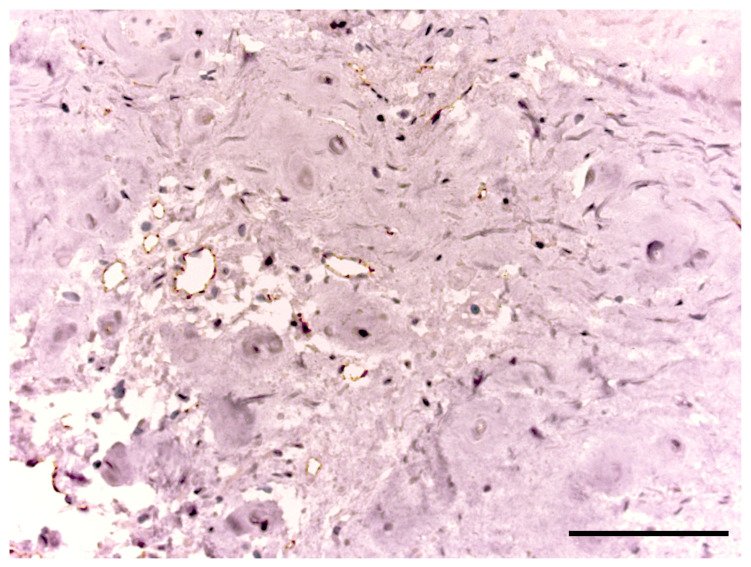
CD31 expression in the intervertebral disc tissue of the patient of the degenerative disc disease group. Scale bar: 100 µm.

**Table 1 biomedicines-11-02184-t001:** Clinical and instrumental indicators in the DDD group, *n* = 34.

Characteristics of Patients	Indicators	MRI Data of the Lumbar Spine	Indicators
Men, women	50%, 50%	Hernia localization, *n* (%)	
		L3/4	–
		L4/5	8 (23.53)
		L5/S1	26 (76.47)
Pain intensity median (VAS, mm), Me [Q1;Q3]	62	The IVD height, mm,	
	[46.75–87.00]	M ± SD	
		at the hernia	0.86 ± 0.16
		level L3/4	2.61 ± 0.15
Variant of the course of pain syndrome, *n* (%)		DDD stage according to Pfirrmann, *n* (%)	
acute	4 (11.76)	3-rd	2 (5.88)
chronic	30 (88,24)	4-th	19 (55.88)
		5-th	13 (38.24)
The nature of pain, *n* (%)		Modic-changes, *n* (%)	19 (55.88)
recurrent	25 (73.53)	Modic-1, *n* (%)	9 (26.47)
persistent	9 (26.47)	Modic-2, *n* (%)	10 (29.41)
Duration of pain syndrome, years	from 0.1 to 25 years	Erosion of the EP, *n* (%)	20 (58.82)
Degree of functional disorders (BP index (BAI)), *n* (%)		Modic + Erosion of the EP, *n* (%)	16 (47.06)
easy	6 (17.65%)		
average	7 (20.59%)		
heavy	21 (61.76%)		

Notes. BP—back pain, IVD—intervertebral disc, DDD—degenerative disc disease, Modic-1—the bone marrow edema, Modic-2—the bone marrow fatty degeneration, and EP—endplates of the vertebral bodies.

**Table 2 biomedicines-11-02184-t002:** Indicators of the expression level of the studied markers in patients determined by the immunohistochemistry.

Marker	Expression Level in Patients, *n* = 34, Average Value %	Expression Level in Controls, *n* = 7, Average Value %
IL-1β	Diffuse in the matrix: 13.06 ± 3.65 (around the vessels 79.24 ± 8.12) Chondrocytes clusters in the NP: 56.15 ± 24.40	Diffuse in the matrix: 1.17 ± 0.47 (around the vessels 4.18 ± 1.72) Chondrocytes in the NP: 16.43 ± 10.10
IL-6	Diffuse in the matrix: 16.33 ± 5.40 Chondrocytes clusters in the NP: 49.65 ± 21.41	Diffuse in the matrix: 3.93 ± 1.17 Chondrocytes in the NP: 29.86 ± 8.45
IL-17	Diffuse in the matrix: 14.15 ± 5.12 (around the vessels 51.89 ± 7.35) Chondrocytes clusters in the NP: 63.71 ± 21.12	Diffuse in the matrix: 0.57 ± 0.35 (around the vessels 3.51 ± 1.60) Chondrocytes in the NP: 8.71 ± 5.77
VEGF-A	Diffuse in the matrix: 16.82 ± 5.70 Chondrocytes clusters in the NP: 74.68 ± 19.17	Diffuse in the matrix: 0.51 ± 0.12 Chondrocytes in the NP: 17.43 ± 7.74
CD31	Diffuse in the matrix: 19.71 ± 9.25	Diffuse in the matrix: 0

Notes. IL—interleukin; VEGF-A—vascular endothelial growth factor A.

## Data Availability

The datasets used and/or analyzed during the current study are available from the corresponding author on reasonable request.

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
