# Peer review of "Immunomorphogenesis in Degenerative Disc Disease: The Role of Proinflammatory Cytokines and Angiogenesis Factors"

_biomedicines, 2023, doi:10.3390/biomedicines11082184_

Round 1

Reviewer 1 Report

In this interesting paper, the authors investigate the cytokines expression profile of a degenerated IVD vs healthy controls (obtained from trauma patients) with an histological and immunoistochemical approach. They found a significant increase in IL-1β, IL-6, IL-17, VEGF-A and CD31 expression with high correlation with the degree of degeneration. They conclude that the condrocytes immunomorphogenic shift into active immune cells is one of the key factors in the initiation and/or maintenance of disc degeneration.

The paper is well written and easy to understand. Figures and tables are essential to comprehension. Auto-citation rate is low (1/27). It fits the scope of the special issue.

For further improvement of the paper, few issues should be addressed:

-Methods: The sequences of the MRI used for the disc evaluation should be specified, along with the protocol of acquisition.

-Methods: No information are given on a previous steroid therapy. Since it is known that steroid deeply affect the immune response also at molecular level, it would be a significant confounding factor. If no data are available, this should be stated into the Limitations section of the paper.

-Carefull check for typo errors (see line 323 for an example)

Author Response

We thank the reviewer for the positive evaluation of our manuscript.

We have revised our manuscript, according to the comments and suggestions.

  1. We’ve specified the sequences of the MRI and updated protocol of acquisition (page 3, 112–117 lines).
  2. We thank for pointing this out. 15% of patients underwent blockade of nerve roots with an anesthetic and hydrocortisone in the period of 30 days before surgery. However, the effect of steroids on disc cytokine levels has not been studied, which is a limitation of the study.
  3. We’ve corrected the typos.

We would like to thank the referee again for taking the time to review our manuscript.

Reviewer 2 Report

The manuscript “Immunomorphogenesis in degenerative disc disease: the role of proinflammatory cytokines and angiogenesis factors” by Pravdyuk et al. investigates the role of proinflammatory cytokines and angiogenesis factors in intervertebral disc (IVD) degeneration and back pain (BP) in young people. The authors performed a clinical examination, MRI, histological and immunohistochemical analyses of IVD tissue samples from 34 patients with BP and DDD who underwent discectomy, and compared them with 7 control samples from healthy individuals. The authors also observed zones of inflammatory infiltration, vascularization, granulation tissue and chondrocyte clusters in the DDD group, indicating an active immunomorphogenesis of chondrocytes and a loss of immune privilege by the disc.

Below are my comments and remarks regarding the manuscript:

1.      Introduction could be improved by providing more background information on the clinical relevance and epidemiology of BP and DDD, especially in young people

2.      The sample size of the study was relatively small (n=34 for the DDD group and n=7 for the control group), which may limit the generalizability and statistical power of the results.

3.      The lack of randomization or matching between the DDD and control groups.

4.      The heterogeneity of the DDD group in terms of pain intensity, duration, nature, location, and functional impairment.

5.      The control group was not matched with the DDD group in terms of age, gender, or other relevant factors, which may introduce confounding variables and bias the comparison.

6.      The possible confounding effects of age, gender, body mass index, smoking status, comorbidities, medication use, and other factors on the expression level of cytokines and angiogenesis markers.

7.      The study did not perform any functional or molecular assays to confirm the causal relationship between cytokine expression and IVD degeneration, or to elucidate the underlying mechanisms of inflammation and angiogenesis in the disc.

8.      The clinical significance and relevance of the results are not clear. The authors do not discuss how their findings could be translated into clinical practice or guidelines for diagnosis, prevention, or management of BP and DDD.

Author Response

We greatly appreciate the thoughtful comments provided on our manuscript. We have carefully addressed all the comments. The corresponding changes and refinements made in the revised paper are summarized in our response below.

  1. We agree and have updated the «Introduction».
  2. This observation is correct. We added it in the «Limitation of Our Study» (pages 12–13).
  3. Our study included patients with a severe stage of DDD (from 3 up to stage 5 according to Pfirrmann), which was dictated by the need to obtain biomaterial as a result of surgery, carried out according to strict clinical indications. Patients with stages 1 and 2 of DDD and a herniated IVD of small size, without critical clinical significance, were not included in the study. There was no sign of a severe stage of DDD in the control group.
  4. More than 60% of patients had severe functional restrictions, 88% – a chronic variant of the course of BP, of which 73% – relapsing course with painless intervals. Half of patients had a 5-year history before the appearance of indications for surgical treatment, which indicates a more vulnerable period in the clinical course of BP from the onset of the disease.
  5. The median age of the control group was 36, which is slightly higher than the mean age of the main group (32 years). We agree that it may somewhat distort the results of the comparison and updated the «Limitation of Our Study» (pages 12–13).
  6. Multivariate analysis of these risk factors, such as age, gender, body mass index, smoking, comorbidity, medication etc. was not included in the objectives of this study and it is a furhter prospect for study. It is now in the «Limitation of Our Study».
  7. The study did not conduct analyzes to confirm a causal relationship. We agree with the remark.
  8. The results of the study can serve as a basis for the use of genetically engineered biological preparations directed against the cytokines IL-1β, -6, -17, VEGF, expressed in the IVD of patients with DDD. We also may try to use biopolymers that prevent neovascularization as a matrix model for intradiscal injection and as a scaffold for stem cell therapy. We added it in the «Conclusions» (page 13).

Round 2

Reviewer 2 Report

ad 7. Maybe at least in the discussion add more potential relationships between cytokine expression and IVD degeneration, or to elucidate the underlying mechanisms of inflammation and angiogenesis in the disc.

Other than that, most of the limitations have been added to the study.
The Editor will decide whether the work is suitable for publication in a small group

Author Response

Thanks once more to the reviewer for providing helpful input on this manuscript.

We have updated the «Discussion» (page 11, 282–300 lines).

We believe that this new version is suitable for publication.